# Decision impact studies, evidence of clinical utility for genomic assays in cancer: A scoping review

Gillian Parker[1], Sarah Hunter[1], Samer Ghazi[2], Robin Z. Hayeems[1,3], Francois Rousseau[4], Fiona A. Miller[1]*

**1** Institute of Health Policy, Management and Evaluation, University of Toronto, Toronto, Ontario, Canada, **2** Lawrence S. Bloomberg Faculty of Nursing, University of Toronto, Toronto, Ontario, Canada, **3** Child Health Evaluative Sciences Program, The Hospital for Sick Children, Toronto, Ontario, Canada, **4** Department of Molecular Biology, Medical Biochemistry, and Pathology, Faculty of Medicine, Université Laval, Québec City, Québec, Canada

* fiona.miller@utoronto.ca

## Abstract

**Data Availability Statement:** All relevant data are within the paper and its Supporting Information files.

### Background

Decision impact studies have become increasingly prevalent in cancer prognostic research in recent years. These studies aim to evaluate the impact of a genomic test on decision-making and appear to be a new form of evidence of clinical utility. The objectives of this review were to identify and characterize decision impact studies in genomic medicine in cancer care and categorize the types of clinical utility outcomes reported.

### Methods

We conducted a search of four databases, Medline, Embase, Scopus and Web of Science, from inception to June 2022. Empirical studies that reported a "decision impact" assessment of a genomic assay on treatment decisions or recommendations for cancer patients were included. We followed scoping review methodology and adapted the Fryback and Thornbury Model to collect and analyze data on clinical utility. The database searches identified 1803 unique articles for title/abstract screening; 269 articles moved to full-text review.

### Results

87 studies met inclusion criteria. All studies were published in the last 12 years with the majority for breast cancer (72%); followed by other cancers (28%) (lung, prostate, colon). Studies reported on the impact of 19 different proprietary (18) and generic (1) assays. Across all four levels of clinical utility, outcomes were reported for 22 discrete measures, including the impact on provider/team decision-making (100%), provider confidence (31%); change in treatment received (46%); patient psychological impacts (17%); and costing or savings impacts (21%). Based on the data synthesis, we created a comprehensive table of outcomes reported for clinical utility.

**Funding:** (FM) Funded by an investigator-initiated grant to FAM from the Canadian Institutes of Health Research (PJT 148805). https://cihr-irsc.gc.ca/e/193.html. The funders had no role in study design, data collection and analysis, decision to publish, or preparation of the manuscript.

**Competing interests:** The authors have declared that no competing interests exist.

**Abbreviations:** PRISMA-ScR, The Preferred Reporting Items for Systematic Reviews and Meta-Analyses statement; HTA, Health technology assessment.

## Conclusions

This scoping review is a first step in understanding the evolution and uses of decision impact studies and their influence on the integration of emerging genomic technologies in cancer care. The results imply that DIS are positioned to provide evidence of clinical utility and impact clinical practice and reimbursement decision-making in cancer care.

**Systematic review registration:** Open Science Framework osf.io/hm3jr.

## Background

Decision impact studies (DIS) propose to evaluate the impact of a medical test or tool on clinical decision-making. Emerging in recent years, DIS appear to be a new form of evidence that has particular relevance to the evaluation of clinical utility, and potential to inform both clinical and reimbursement decision-making for genomic technologies used in cancer care. Though new, DIS have already been referenced in numerous international clinical practice guidelines and used to inform reimbursement decisions for genomic assays in collectively financed health systems [1–4]. As new types of evidence are developed and disseminated within the field, it is the responsibility of researchers to interrogate these sources of evidence, in order to understand their place within decision-making for clinical practice, coverage and/or reimbursement.

Genomic assays are emerging technologies increasingly used in cancer care [5, 6]. These assays are algorithm-based tools that examine multiple gene sequences of a tumor to assess prognosis, and in some cases predict response to treatment for a patient [5, 7]. The results can identify patients who will most likely respond to a specific therapy (e.g., adjuvant chemotherapy) based on the stratification of the probability of a clinical outcome [8]. One potential benefit of genomic assays is the potential to avoid "overtreatment" or recommend intensification of treatment based on a tumour profile. Currently, genomic assays are most commonly used in breast cancer prognostics, but are increasingly being developed for other cancers, such as prostate, colon and lung cancers [8]. Most prognostic assays are proprietary and are developed by commercial entities [6, 9]. For example, the top five genomic breast cancer prognostics are all proprietary products: Oncotype Dx (Exact Sciences), followed by Mammaprint (Agendia), Prosigna (Nanostring), Breast Cancer Index (Biothernostics) and Endopredict (Myriad). For other cancers, Envisia (Veracyte) and Percepta (Vercyte) for lung cancer and Decipher (Scipher) for prostate cancer are the most common and proprietary products. Exact Sciences also offers genomic assays for ductal carcinoma in situ (DCIS) breast, prostate and colon cancers. The prevalence of industry in the development and delivery of these assays may be significant in the creation and proliferation of decision impact studies.

Critics note that while analytical validity (ability to detect the analyte) and clinical validity (ability of the analyte to detect a clinical phenomenon) may have been established for these assays, clinical implementation has been limited due to a lack of evidence of clinical utility (utility of clinically valid analyte results) [8, 10, 11]. Establishing clinical utility is particularly challenging in an emerging field like genomic medicine because producing direct evidence that the use of an assay will result in a net improvement in the patient's condition is a costly and time-consuming endeavour [11]. As with all diagnostics or prognostics, these genomic assays do not directly act on health outcomes; instead, they inform decision-making about risk profiles or the use of therapeutic interventions. Efforts to measure the clinical and economic value of a test must therefore consider a "chain of evidence" linking intermediate to ultimate

outcomes [10, 12]. The links in this chain typically assess the analytic validity, clinical validity and clinical utility of the test, with the final step, clinical utility, defined as something that improves patient outcomes and adds value to the clinical decision-making process [10]. Clinical utility is viewed as a key standard for reimbursement decision-making for healthcare interventions, including diagnostic and prognostics tools [7, 9, 11, 13–18]. While the concept of clinical utility is epidemiologically clear, relating to a test's demonstrated clinical effectiveness [13], in practice there is considerable variation in what is accepted as sufficient proof [7, 13, 16, 17]. Indeed, increasingly, the definition is being loosened to incorporate broader conceptualizations of value [11, 13, 17, 19], including both direct and indirect indicators of patient outcomes, health outcomes as well as non-health outcomes, and societal effects that relate to family impacts, societal acceptability and value for money [16].

A lack of evidence of clinical utility for genomic assays is a recognized challenge in the field. A 2015 synthesis of systematic reviews of the clinical utility of gene-expression profiling in breast cancer reported that the included studies "form part of the evidence base on the potential impact of the clinical use of [genomic assays]" [5 (p.520)]. In the absence of direct evidence, the field appears to be relying on multiple outcomes reported as evidence of clinical utility. While three large-scale, prospective randomized trials assessing breast cancer prognostics assays (OncotypeDx—TAILORx, RxPONDER; Mammaprint–MINDACT) have reported beneficial patient outcomes as a result of using the genomic assay in recent years [20–22], clinical trials are expensive and can take 5 to 10 years to produce results [8] and may not provide jurisdiction specific data for reimbursement decision-making. In addition, payers may be resistant to reimburse a test without established clinical utility, which presents significant challenges for diagnostic companies [11]. Decision impact studies may be positioned as an intermediary resource to provide more timely, less costly and jurisdictional-specific evidence of clinical utility for reimbursement. Understanding the reimbursement landscape for genomic assays provides context for understanding the development and propagation of decision impact studies. Over the last 10 years some of the genomic assays have gained reimbursement in multiple jurisdictions, but even established assays are still seeking reimbursement in many countries and for various patient populations [6]. Numerous international health technology assessment (HTA) evaluations of genomic breast cancer prognostics have assessed the clinical utility of these assays with uneven results [7]. For example, the French health authority, Haute Autorité de Santé (HAS), issued a report in early 2019 on the lack of evidence in favour of genomic tests, which prevented full reimbursement of these assays by national health insurance [23]. In 2019, the Medical Services Advisory Committee (MSAC), the Australian health reimbursement authority, reported that the major issue in previous submissions for four proprietary breast cancer prognostic assays was that comparative clinical utility had not been demonstrated in Australia (or elsewhere). The Committee stated that substantial uncertainty remains about the relative analytic performance, clinical validity and especially clinical utility of these assays in the Australian context [3]. Conversely, several proprietary breast cancer prognostics have been approved for coverage by Medicare and Medicaid in the US since the mid-2000s [24]. Also, in 2018, the UK National Institute for Health and Care Excellence (NICE) recommended EndoPredict, Oncotype Dx and Prosigna as options for guiding adjuvant chemotherapy decisions for specific breast cancers and included 'decision impact' as a category for assessing clinical effectiveness. The varied decision outcomes of these reimbursement decision-making processes provide insight into the centrality of clinical utility to reimbursement decision-making.

## Study objectives

To date, no reviews or meta-analyzes of decision impact studies have been published, which limits knowledge about the objectives and outcomes of these studies. Therefore, there is a need

for a systematic examination of this literature to identify and characterize decision impact studies in genomic cancer testing. The secondary objective of this review was to categorize the types of clinical utility outcomes reported as evidence to begin to understand the creation and intended role of DIS in clinical and reimbursement decision-making.

### Identifying the research questions

Through an iterative process and based on the results of a preliminary literature review, the following research questions were developed:

RQ1: What are the characteristics of published decision impact studies?

RQ2: What types of cancer research and genomic tests/assays use decision impact studies?

RQ3: What outcomes do decision impact studies report and how do reported outcomes align with existing measures of clinical utility?

## Methods

A scoping review is a useful methodology to determine the coverage of a body of literature on a given topic and to identify and analyze knowledge gaps [25]. We used Arksey and O'Malley's framework for scoping reviews and incorporated enhancements by Levac et al., [26]. The Preferred Reporting Items for Systematic Reviews and Meta-Analyses statement (PRISMA-ScR) was used to guide the reporting process [27] (see S1 Appendix). An early protocol for this study was registered with Open Science Framework osf.io/hm3jr.

### Identifying relevant studies

We conducted a comprehensive search of four databases, Medline, Embase, Scopus, and Web of Science, including publications from the inception of each database to June 2022. Only empirical studies, both articles and conference abstracts, were included. Scope was limited to empirical studies to ensure all included items reported outcomes based on verifiable evidence. The rationale for including conference abstracts was to track the origins and growth of these studies. As we are exploring the production of 'scientific' evidence, only publications listed in the selected databases, and not in the gray literature, were included. Due to resource constraints, only English language studies were included in this review. As we are interested in this specific type of study, for the purposes of this review we define a decision impact study as an empirical study that references its primary objective as a "decision impact" assessment. Specifically, the primary outcome is the impact of a genomic assay on treatment decisions or recommendations for a specific population of cancer patients.

### Search strategy

We conducted a focussed search for studies that used the exact phrases "decision impact" or "decision-impact" or "decision-making impact" or "decision making impact" without limitation of other search terms (see S2 Appendix). Our plan with this broad search was to ensure we captured all decision impact studies, with the intention of screening for DIS in cancer and genomics at the title/abstract screening phase.

### Study selection

Database search results were imported into Covidence, a Cochrane technology platform, (www.covidence.org) to facilitate the screening of the article titles and abstracts. The title and abstract screening process was conducted by three research team members (GP, SG, SH). Two

reviewers (SH and SG) screened all titles and abstracts independently. One research team member (GP) reviewed a random 10% sample of screened abstracts and resolved discrepancies. As mentioned above, during title and abstract review, studies that were not related to cancer or genomics were excluded. Only empirical studies focussed on use of genomic assays in cancer care were moved to full-text review. Full-text review was conducted by two reviewers (SH and SG), with the third reviewer (GP) checking a random 10% sample of articles to ensure reliability. Discrepancies were discussed and resolved collaboratively. Conference abstracts were excluded for studies that published their full results in articles.

## Data collection and extraction

The data collection worksheet was designed iteratively. It was piloted with 20 studies that met the eligibility criteria and was revised based on the results of the pilot. Data extraction worksheets are used in scoping reviews to provide a structured and detailed summary of each study and were used to identify and organize information on included items. Data were collected on characteristics of DIS including publication details, type of cancer/disease, geographic study setting, study design, and operationalizations of clinical utility.

The Fryback and Thornbury hierarchical model of efficacy (FT Model) is an evaluative framework used to support the assessment of diagnostic imaging tests; it has also been used in other areas of diagnostics. It offers a comprehensive set of domains of efficacy that map onto the concept of clinical utility. The largely hierarchical and nested nature of the framework is well-suited to the context of genomics because the components of effectiveness are specific, well defined, and linked as a chain of evidence [10, 16]. The Model consists of six levels of efficacy, with levels 3–6 pertaining to clinical utility (see Table 1).

We used levels 3–6 of the FT Model [28], with recent adaptations [10, 16] and further modifications based on the results of our pilot test to collect clinical utility data.

## Analyzing the data

The data were entered into an Excel spreadsheet version of our data collection worksheet, for analysis and reporting. Descriptive statistics were used to summarize the data by categories. The data were analyzed by three members of the research team (SG, SH and GP) with discrepancies resolved collaboratively. The FT Model [28] and adapted versions [10, 16] were used to analyze the data relevant to clinical utility. All members of the research team reviewed the final summary of findings.

## Results

### Literature search

The database searches identified 1803 articles (after duplicates were removed) for which the titles and abstracts were screened for inclusion. Of these, 269 articles were selected for full-text

**Table 1. Fryback and Thornbury hierarchical model of efficacy (FT Model).**

| Value | Level | Name | Indicator |
|---|---|---|---|
| Analytical Validity | 1 | Technical efficacy | *i.e., laboratory performance* |
| Clinical Validity | 2 | Diagnostic accuracy efficacy | *i.e., clinical sensitivity and specificity* |
| **Clinical Utility** | **3** | **Diagnostic thinking efficacy** | *i.e., impact on clinician's diagnostic process* |
| | **4** | **Therapeutic efficacy** | *i.e., impact on clinical management* |
| | **5** | **Patient outcome efficacy** | *i.e., patient benefit* |
| | **6** | **Societal outcome efficacy** | *i.e., cost-benefit, cost effectiveness, societal acceptability* |

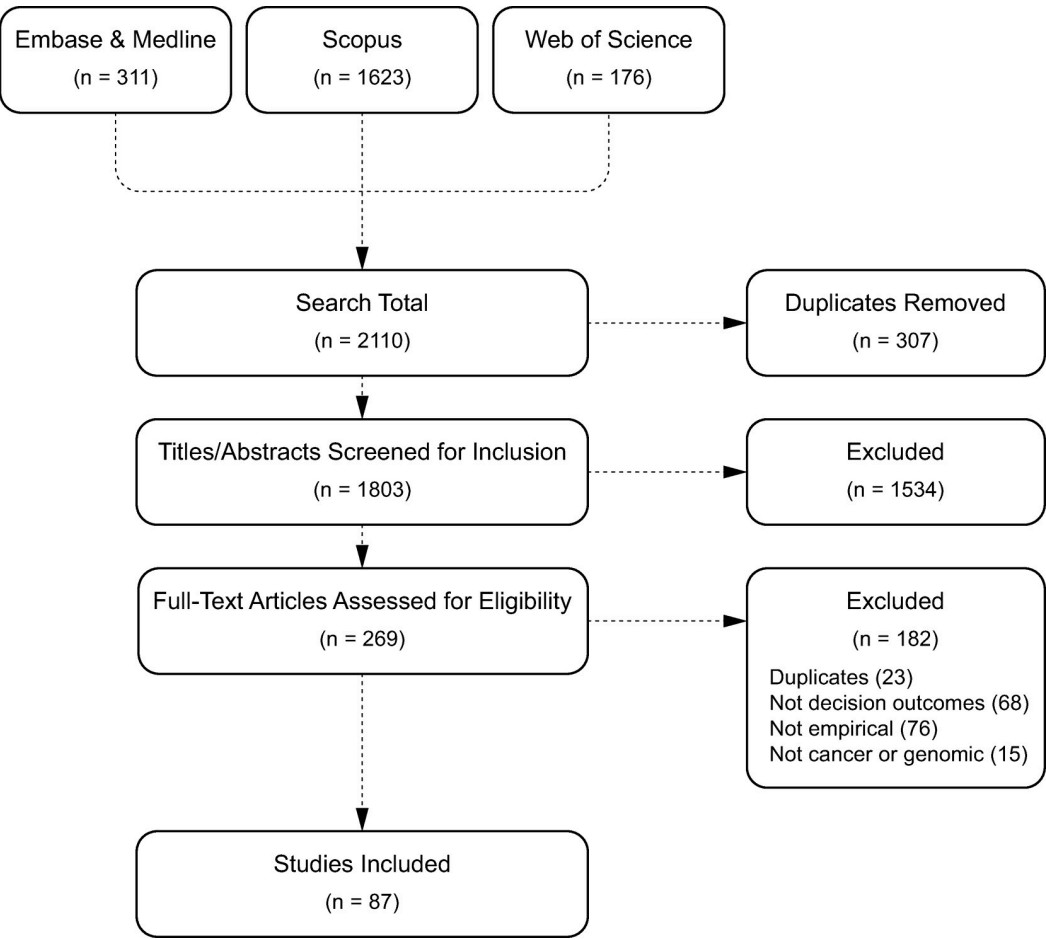

**Fig 1. PRISMA diagram of study selection process.**

screening and 87 studies (46 articles and 41 conference abstracts) [29–115] were included in this review. See S1 Table for table of included studies. See Fig 1 for the PRISMA diagram representing the complete article selection process.

## Study characteristics

**Timeline.**   While databases were searched from inception, the first DIS was published in 2011 with the first seven publications (2011–2012) being conference abstracts. Sixty-seven percent of the included studies have been published since 2016. Decision impact studies were initially conducted only for genomics assays used in breast cancer care, but in 2013, publications on genomic assays used in lung, pancreaticobiliary, prostate and unknown cancer care started to emerge. Fig 2 demonstrates the publication of DIS over time.

**Cancer- and genomic assay-type.**   The vast majority of studies were related to breast cancer (n = 63), followed by other cancers (n = 24) (e.g., lung, prostate and colon). Fig 3 shows the distribution of studies by cancer-type. The included studies examined the impact of 19 different genomic assays, predominately proprietary (n = 18), with only one study assessing a generic assay. Oncotype Dx (Genomic Health/Exact Sciences) for breast, DCIS breast and colon cancers were researched in half of the included studies (n = 44), followed by Prosigna (Veracyte) for breast cancer (n = 9), Percepta (Veracyte) for lung cancer (n = 5),

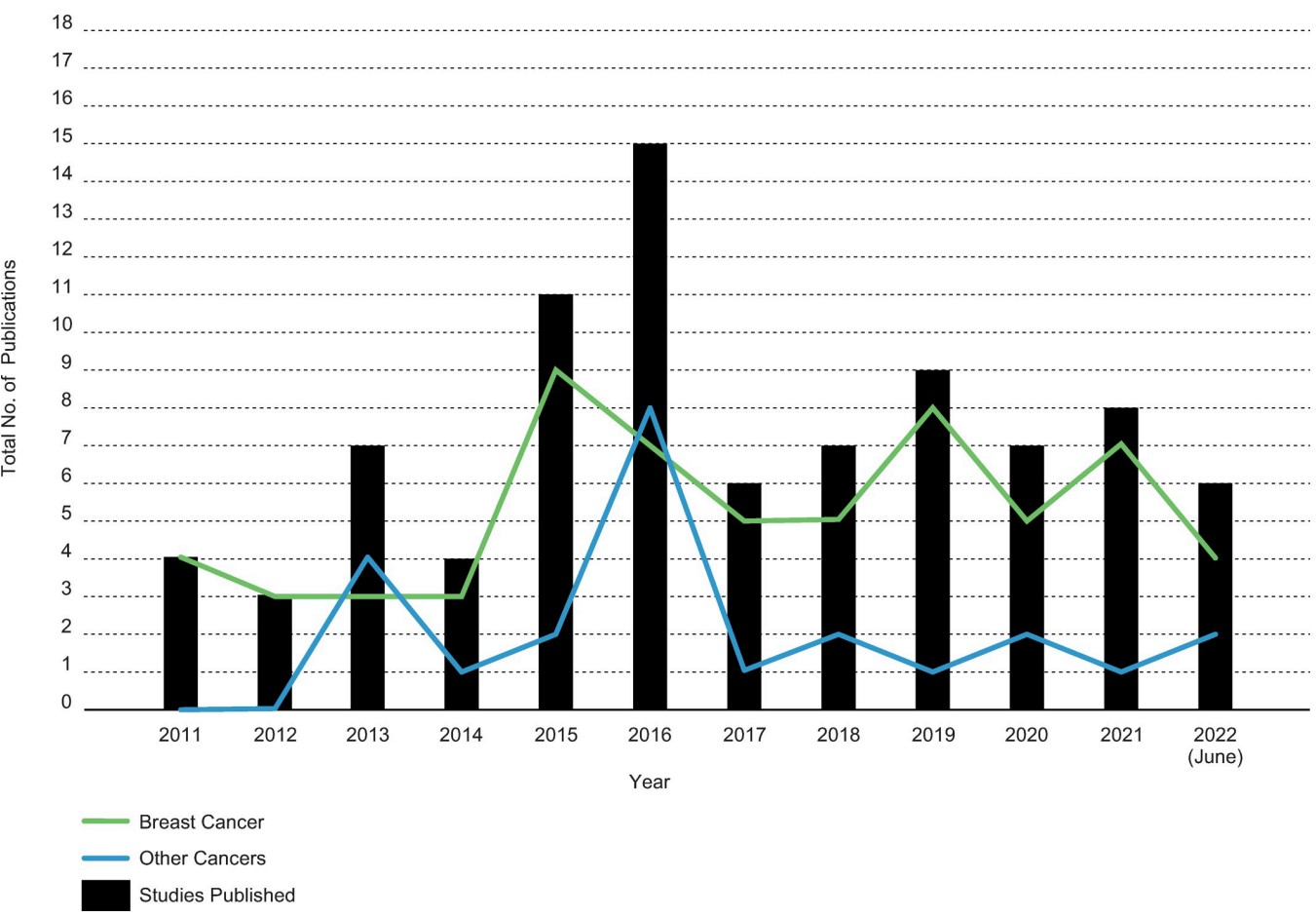

**Fig 2. Decision impact studies publication over time.**

FoundationOne (Foundation Medicine) for various cancers (n = 4) and Endopredict (Myriad Genetics) for breast cancer (n = 4) assays.

**Geography.** The included studies were conducted in 24 countries. The majority were in Europe (n = 33); followed by North America (n = 23); the Middle East (n = 9), South America (n = 6), Asia (n = 6), Australia (n = 5) and the United Kingdom (n = 5). While breast cancer represented 72% of included studies (n = 63), only 6% (n = 5) of these studies were conducted in the US. In addition, 3 of 5 of the US breast cancer studies were conducted by the same study team for the Breast Cancer Index tool. Breast cancer studies were most frequently conducted in Germany (n = 9) and France (n = 9). The majority of US studies (i.e., 76%) were related to other cancers, such as lung cancer (n = 6) or prostate cancer (n = 4).

## Study design

The majority of studies used a prospective study design (n = 65), followed by a retrospective design (n = 19); three studies used a prospective and retrospective design. The prospective studies used a pre- post- survey/questionnaire data collection method to collect data on change in decisions and/or treatment, retrospective studies primarily used a retrospective sample or chart analysis. The majority of decision-makers in the included studies were physicians

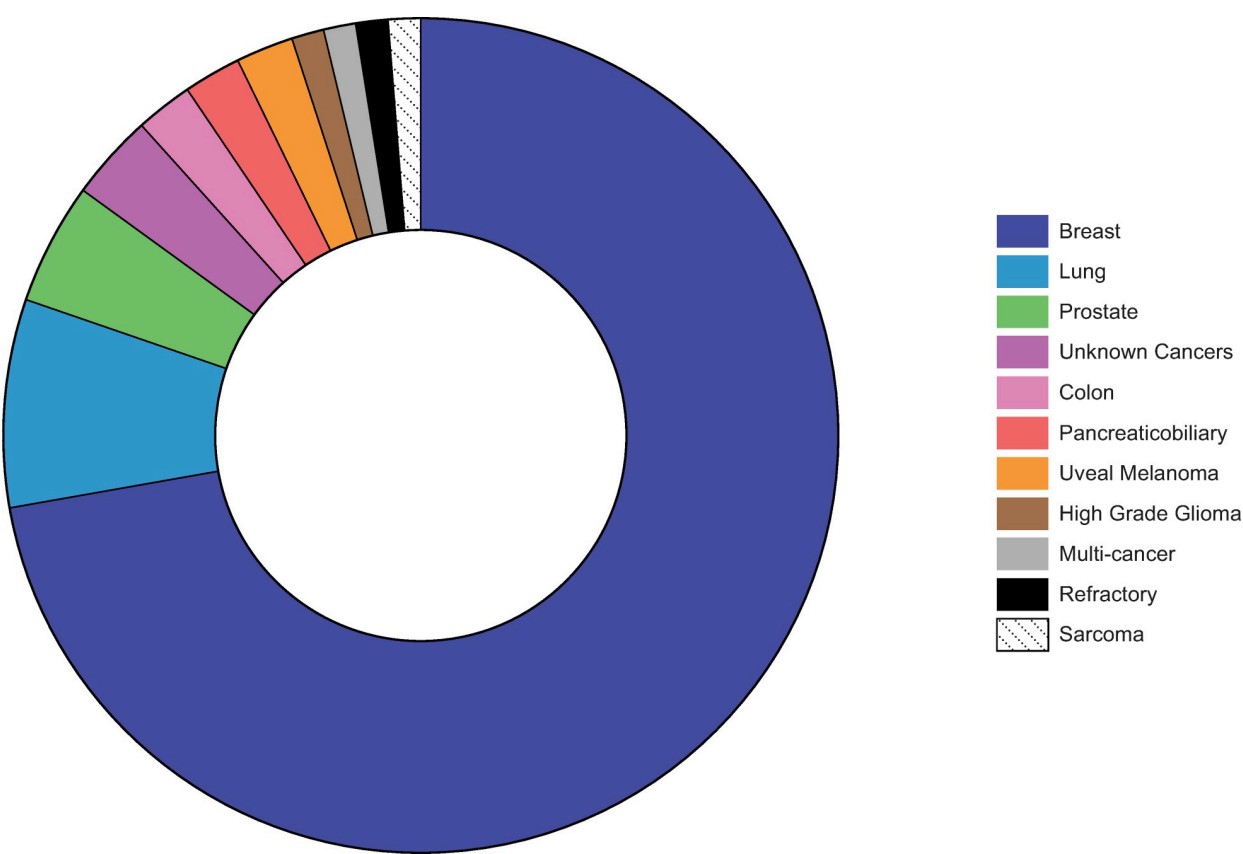

**Fig 3. Distribution of cancer types.**

(n = 72), followed by tumor board/ multi-disciplinary teams (n = 15). Ten studies reported on both provider and patient decision-making.

## Decision impact study label

All included studies used the term "decision impact". A third (n = 30) of the included studies used "decision impact" in the publication title. Use of "decision impact" as an author's keyword began in 2013, and 17 studies used the keyword. Numerous studies (n = 14) described their research as "the first" decision impact study in a jurisdiction, for example: "...we published one of the first and largest real-world decision impact studies"[76 (p790)]; "This is the first decision impact study to include..." [102 (p771)]; "This prospective decision impact study of the 21-gene breast cancer assay is the largest to date in Latin America and the first such study within the Mexican public health care system." [34 (p205)]

## Clinical utility and decision impact studies

We identified 22 discrete measures/indicators that aligned with the clinical utility levels of the FT Model. Table 2 demonstrates the reported clinical utility outcomes categorized under FT Model levels and definitions, measurement constructs, sample measures/indicators, and examples from the included studies.

**Table 2. Table of reported clinical utility outcomes in DIS assessing the impact of genomic assays in cancer.**

| FT Model domain & definition | Measurement construct | Measure/Indicator | Example from included studies |
|---|---|---|---|
| **LEVEL 3: DIAGNOSTIC THINKING EFFICACY**<br>*Does the test impact the prognostic/ diagnostic assessment?* | Impact on clinical recommendation(s): % Patients that received a modified prognostic or diagnostic assessment based on genomic test results | Change in clinical recommendations | *Y/N* |
| | | REDUCTION in recommendation for unnecessary treatment | *"Post-21-gene testing, 602 patients (62.5%) experienced a change in treatment recommendation: 593 (61%) avoiding adjuvant chemotherapy"* |
| | | ADDITION of recommendation for beneficial treatment | *"Recommendation for antifibrotic treatment increased from 15 (10%) pre-Envisia to 72 (46.4%) post-Envisia, while the recommendation for SLB or Cryo decreased with Envisia (OR = 0.48, p = 0.03)."* |
| | | NET CHANGE of recommendation for treatment | *"When the urologists did have knowledge of the GC test results, treatment recommendations changed by 31% from when they were without knowledge of GC (95% CI 27–35%)"* |
| | | Recommendation for change in treatment intensity | *". . .resulting in a decrease in treatment intensity in 76 patients (28.2%); and an increase in treatment intensity in 26 patients (9.7%)"* |
| | Impact on patient decision-making | Patient decision to follow or not follow recommendation(s) | *"Twenty-one (9%) N0 patients did not follow their physician's post-test recommendation. . . Twenty-four (20%) N+ patients decided against their physician's post-test recommendation."* |
| | Impact on physician confidence | Physician confidence increase or decrease based on genomic test results | *"Before testing, physicians reported feeling somewhat confident (78%) or strongly confident (6%) in 84% of their treatment recommendations. After testing, physicians reported feeling strongly confident in 99% of their treatment recommendations."* |
| **LEVEL 4: THERAPEUTIC EFFICACY**<br>*Does the test change or cancel planned treatment?* | Impact on intervention(s) | Change in intervention | *Y/N* |
| | Prevention and treatment optimization outcomes | REDUCTION in actual use of unnecessary treatment | *"Extended therapy was recommended for 75% patients pre- and for 55% post-testing."* |
| | | ADDITION of actual use of beneficial treatment | *"In patients with high RS values, there was a 27% relative increase of actual chemotherapy use."* |
| | | NET CHANGE in actual unnecessary treatment | *"Of the 24 patients who were recommended endocrine therapy before testing, 1 patient (4.2%; 95% CI 0–22.7%) received chemoendocrine therapy after testing; of the 23 patients who were recommended chemoendocrine therapy before testing, more than half (12 patients, 52.2%; 95% CI 31.4–72.3%) received endocrine therapy alone after testing."* |
| | | CHANGE in actual treatment intensity | *". . .knowing the Recurrence Score result led the treating physician to intensify the chemoendocrine therapy plan by adding taxane-based regimens after the originally planned doxorubicin/ cyclophosphamide (adriamycin and cyclophosphamide) treatment."* |
| **FT Model domain & definition** | **Measurement construct** | **Measure/ Indicator** | **Example from included studies** |

*(Continued)*

**Table 2.** (Continued)

| | | | |
|---|---|---|---|
| **LEVEL 5: PATIENT OUTCOME EFFICACY** *Do patients who take the test fare better than similar patient who do not?* | Health-related | General: clinical response rate, life years gained, adverse event rate | *"The 21-gene assay was projected to increase the mean life expectancy by 0.06 years and the quality-adjusted life expectancy by 0.06 quality-adjusted life years (QALYs) compared with current clinical practice over a 30-year time horizon."* |
| | | QOL-related: Quality of life years (QALY) | *"By reducing the chemotherapy disutility, and preventing recurrence, the RS increased the patient's quality adjusted life (QALYS) by 0.215 years."* |
| | Non-health-related | Psychological: Decisional Conflict Scale | *"Decision conflict significantly decreased following BCI testing (44.8–36.3; p < 0.0001)."* |
| | | Psychological: State Anxiety | *"After the post-Prosigna test treatment recommendations, patients' STAI component of state anxiety was statistically significantly decreased post-Prosigna test relative to pre-Prosigna test (p = 0.02) (S1 Table)."* |
| | | Psychological: Trait Anxiety | *"Overall, patients' Trait Anxiety scores remained virtually unchanged (increase of 0.008, [− 0.81 to 0.82]). In high-risk patients, there was a modest but significant (p = 0.02) average increase of 1.76 [0.26 to 3.26] in Trait Anxiety scores. In low-risk patients, there was a slight (p = 0.06, borderline significant) decrease in Trait Anxiety: the mean change was − 0.90 points [− 1.86 to + 0.04]."* |
| | | Psychological: Other tool(s) | *"The same significant association was observed with the emotional well-being subscale (p<0.05) and the functional well-being subscale (p<0.01) of the FACT-G, v.4. Prosigna test results had a positive effect (improving emotional and functional well-being) for patients categorized as ROR high risk."* |
| **LEVEL 6: SOCIETAL OUTCOME EFFICACY** *Do cost benefit or cost-effectiveness analyses indicate that the test has efficacy at the health system or societal level?* | Cost of testing | Cost of genomic assay per patient or overall | *"Genomic tests are reimbursed 1,849.50€."* |
| | Cost of adjuvant treatment (actual or est.) | Cost savings or increase of adjuvant treatment per patient or overall | *"Testing with the 21-gene Recurrence Score cost ¥350,000 per patient with ¥153,490 lower acute costs because fewer women were treated with adjuvant chemotherapy."* |
| | Net cost savings or increase (actual or est.) | Mean annualized savings rate, net savings rate, average cost savings per patient | *"Deducting the assay cost, net savings of over one million euro was achieved."* |
| | Cost-effectiveness | Cost per QALY gained, incremental cost effectiveness ratio (ICER) | *"Assessment of cost-effectiveness showed that the use of Oncotype DX was associated with an incremental cost-effectiveness ratio (ICER) of GBP 6232 per QALY gained and GBP 5633 per life year gained in comparison with current clinical practice."* |

Table 3 below shows the number of studies with reported outcomes for each of the clinical utility levels of the FT Model. All studies reported outcomes that mapped to *Diagnostic Thinking Efficacy* as the impact on decision-making is the purpose of this type of study.

Four studies reported outcomes that mapped to all four FT Model levels and 19 studies reported outcomes that mapped to three different levels. Fig 4 demonstrates the distribution of outcomes by FT Model level by study.

*Diagnostic Thinking Efficacy* outcomes report on the impact of the test results on the thinking of the clinician who ordered the test; this was proposed as an intermediate step linking the

**Table 3. Number of studies reporting outcomes per FT Model level.**

| Level | Name | Number of studies reporting outcomes |
|---|---|---|
| 3 | Diagnostic Thinking Efficacy | 87 |
| 4 | Therapeutic Efficacy | 40 |
| 5 | Patient Outcome Efficacy | 22 |
| 6 | Societal Outcome Efficacy | 18 |

information in the test results to changes in the treatment of the patient. 77 studies reported a net change in decision or recommendation, with 76 studies reporting a decision to reduce unnecessary treatment and 49 studies reporting a decision to add beneficial treatment. In addition, approximately one third (n = 27) of studies reported on physician confidence, with all reporting that physician confidence increased as a result of using the assay. The majority of studies used a questionnaire with Likert scale questions; two studies referenced a scale used in a previous DIS, and two studies used the Decisional Conflict Scale. These outcomes are illustrated in Sethi et al.'s DIS of Percepta Genomic Sequencing Classifier for patients with high-risk lung nodules, where pulmonologists were surveyed to assess the impact of the test results on their confidence in treatment recommendations [95]. The authors reported that the results provided by the genomic sequencing classifier increased provider confidence, with 76% of survey takers rating their level of confidence as high after evaluating the test results. Patient decision-making was reported in 10 studies. These studies reported that the majority of patients decided to follow the recommendations based on the results of the genomic assay. In their Canadian DIS of Oncotype Dx for ER-positive, node-positive patients with breast cancer, Torres and colleagues reported that 53% of patients changed their treatment preference after receiving the test result [102].

*Therapeutic Efficacy* captures outcomes related to a change to an *a priori* treatment plan based on results of the genomic assay, such as recommending against adjuvant chemotherapy. As stated, all of the included studies reported a change in decision but only 46% (n = 40) reported an actual change in treatment. Treatments impacted by these decisions were primarily chemotherapy (n = 37), followed by other treatments (n = 6) (e.g., radiation, surgery,

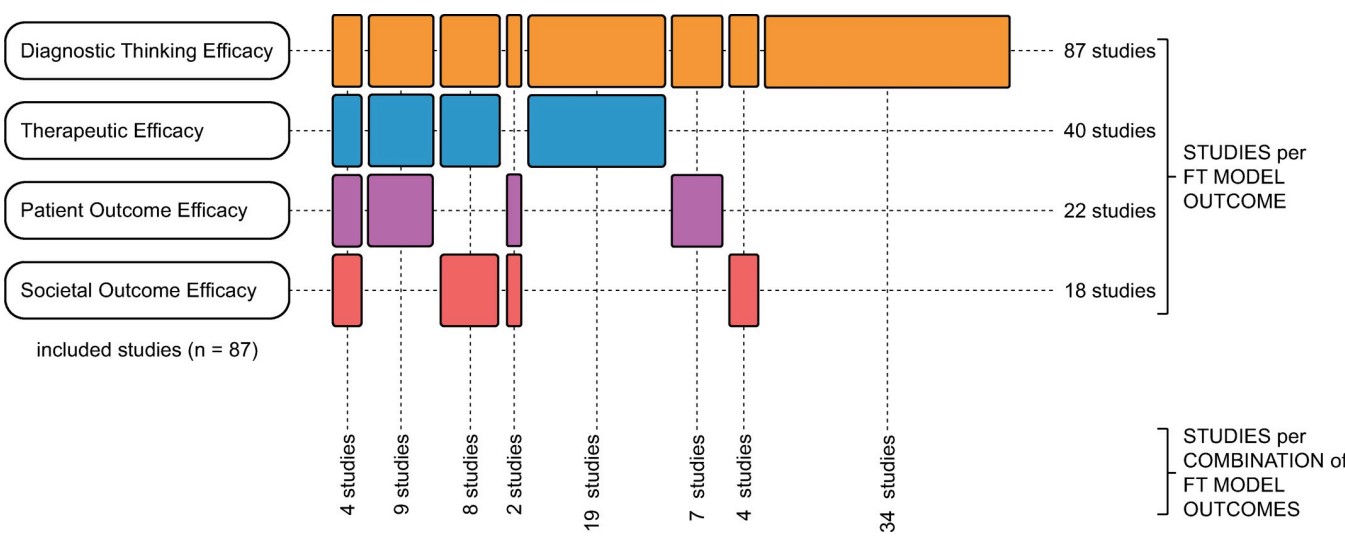

**Fig 4. Reported outcomes against FT Model levels.**

endocrine therapy). For example, Hequet et al., in their DIS of the impact of Prosigna for HR +, HER2- BC patients, provided detailed outcomes related to change in treatment as they reported both change in recommendation and actual treatment [64]. The authors reported that they collected data on treatment change for 79% of their participants and that the recommendation based on the assay results was applied in 85% of cases [64].

The primary patient related outcomes reported that aligned with *Patient Outcome Efficacy* were regarding patient psychological assessments. Of the 15 studies that conducted a patient psychological assessment, all used validated scales (Decisional Conflict Scale, State/Trait Anxiety Test). Five studies reported on other psychological assessments, such as the Functional Assessment of Cancer Therapy–General and patient confidence. All studies that included a psychological assessment (n = 15) reported that patients' decisional conflict decreased, confidence increased, or anxiety/worry decreased as a result of the use of the test/tool. Forty percent of these studies included other patient related outcomes, such as clinical response rate, life-years gained, or adverse event rate results. A few studies (n = 3) reported on an increase in Quality-of-Life years (QALYs). Holt and colleagues published a comprehensive, early decision impact study in 2013 focussed on the use of Oncotype Dx for ER-positive, node-negative breast cancer patients. In their study, they not only reported on change in decision, but also on patient confidence and quality-adjusted life-year (QALYS). The study reported that patient's decision conflict decreased, and confidence increased as a result of the genomic assay results.

Outcomes related to *Societal Outcome Efficacy* report on the extent to which the test is an efficient use of societal resources to provide medical benefits to society. These types of outcomes were reported in 18 DIS. Thirteen studies reported on net costs or savings (cost of the genomic assay against the cost savings of reducing adjuvant treatment based on test results) and four studies reported on cost effectiveness through either cost per QALY gained or incremental cost effectiveness ratio (ICER). As the focus of this review was decision impact studies, the included studies do not represent all research into costs and economic analyses of this field, but it is important to document the inclusion of these clinical utility outcomes in this new type of evidence.

## Discussion

Decision impact studies are an emerging form of evidence in the growing field of genomic medicine. This body of research has developed in a short period of time and appears to be a purposive creation to meet a specific need. Clinical utility definitions are contested, and robust clinical outcomes are currently underdeveloped for genomic assays in cancer care. The findings of this review demonstrate that these studies are clearly focused on reporting outcomes of clinical utility which is a key issue for reimbursement decision-making. Understanding decision impact studies as a new type of evidence is valuable, particularly as the findings suggest that these studies are being conducted to provide surrogate evidence of clinical utility.

This review identified 87 decision impact studies published in the last twelve years. The number of these publications has increased significantly since the initial set of conference abstracts published in 2011. Study publications peaked in 2015 for breast cancer and 2016 for other cancers, which appears to align with key reimbursement decision-making in the US and Europe though the relationships between publications, HTA evaluations and reimbursement decision-making requires further investigation. Our results also show that decision impact studies started in breast cancer research and subsequently spread to other types of cancer research. Breast cancer is a heterogeneous disease; for most women with breast cancer, the cure rate is high [5]. These studies may have proliferated in breast cancer research because, for a subset of the remaining women, genomic assays may be able to play a role in identifying

risks and facilitating the most appropriate treatment plan [8]. In contrast, the use of genomic assays in other fatal diseases may have limited clinical utility since most patients with poor prognosis cancers will already receive the most intensive treatments; therefore, a decision aid, like a genomic test result, may add little value [8].

The included studies were conducted primarily for proprietary assays in numerous countries in Europe, North America, the Middle East and Asia. Breast cancer prognostics were the focus of the majority of these studies, and this is likely to do with Genomic Health's (the US based producer of Oncotype Dx genomic assay) place at the forefront of genomic breast cancer prognostic assays. As previously illustrated, since the mid-2000s, these assays have been covered by Medicare and Medicaid in the US. While Oncotype Dx studies are the majority of included studies, there are no DIS for Oncotype Dx conducted in a US setting in our sample. Instead, the studies of genomic breast cancer prognostic were conducted primarily in countries such as Australia, Mexico, Turkey, Canada and France, where reimbursement is being sought, has been denied or restricted. These results give weight to the suggestion that jurisdiction specific studies are conducted to support reimbursement processes and local requirements for evidence of clinical utility.

The label "decision impact study" began in 2010 and the use of the keyword "decision impact" began in 2013. Of note, many of the included studies labeled themselves as a "decision impact study" and numerous studies described their research as "the first" decision impact study in a jurisdiction. This positioning may be intended to validate this new type of research and present the study as a known, standardized type of research. As well, the included studies primarily used a prospective design, which may be a response to early critics, who questioned the heavy reliance on retrospective studies in the field [9, 116].

Outcomes for clinical utility were reported across 22 discrete measures/indicators, corresponding clearly to the four levels of clinical utility identified by the FT model. By definition, all included studies reported a change in decision, but less than 50 percent of studies reported on a change in actual treatment, raising questions about which outcomes are most important from the perspective of judging clinical utility. This result aligns with the critique is that results of this sort are presented in the absence of consensus on the relative value of the different outcomes. Without direct patient outcomes, surrogate outcomes are used in the chain of evidence as demonstrated by the results reported in the decision impact studies. The reporting of "change in decision" as opposed to (or in addition to) "change in treatment" should be interrogated as a surrogate outcome. If improved patient outcomes are the end goal, changes to actual treatment would appear to offer the most value. While the validity of, or value added provided by reporting a change in decision has been debated, Frybach and Thornbury state that test results may change the course of treatment, or they may just reassure the physician [28]. The authors contend that providers see value in results that only reassure them regarding an *a priori* diagnosis [28]. Further research must be done to understand the importance placed on specific outcomes and how these clinical utility outcomes are utilized in clinical and reimbursement decision-making.

Over one third of the studies in our review evaluated physician confidence as a component of clinical utility. The role of provider confidence in clinical utility is debated, but features prominently in these studies. Frybach & Thornbury discuss the value of reassurance, a version of confidence, in their model [28]. Walcott et al. found that the diagnostic thinking efficacy level was not prevalent in their review of the literature [10]. The authors acknowledge the importance of measuring "the extent to which a test result helps a clinician come to a diagnosis and/or how the test results compare to a clinician's pretest estimate of the probability of disease" [10 (p384)] and call for future work to explore measures of diagnostic thinking efficacy.

Studies reported on psychological assessments of the impact of the test results on patients' decision conflict, stress, or anxiety. These assessments are common in providing data to understand the positive or negative impact on patients' mental health regarding results. Patient confidence is typically categorized under non-health-related outcomes and broader definitions of value outside of traditional definitions of clinical utility [16, 19]. It is important to note that patient psychological assessments were only reported in breast cancer studies, not for other cancers. This result may be a reflection of the fact that decision impact studies started and are most prominent for breast cancer prognostics.

Evaluations of costs, savings and cost-effectiveness are particularly relevant because the majority of assays studied in this review were proprietary products which are often expensive for reimbursors. For the decision impact studies that included cost analyses, the cost of the assay (which is often expensive) is presented against the potential or actual savings from adjuvant chemotherapy treatments not used. The majority of studies that included outcomes related to costs reported that, while expensive, the downstream savings were more than the cost of the assay for the payer. Costs and financial outcomes are critical for reimbursement decision-making and this aspect of decision impact studies requires further investigation.

As clinical utility is one of the key evaluation criteria for HTAs used to determine coverage and reimbursement for diagnostics and prognostics [16, 20, 117, 118], it is important to understand the role of DIS in these assessments. The varied requirements for clinical utility and lack of clear evidentiary requirements appear to have supported the creation and proliferation of DIS as a surrogate outcome to provide links in the chain of evidence.

### Directions for future research

Understanding the intended purpose and goals of DIS is critical to situating them in the context of decision-making for clinical practice, coverage and reimbursement. While this review provides much foundational information, it also illuminates important questions for further inquiry in the field:

- The included studies reported multiple outcomes against clinical utility—how many outcomes are needed to fulfill the chain of evidence? What would constitute "enough" evidence to provide robust evidence of clinical utility?

- How does industry involvement in research of proprietary products impact usage, coverage and reimbursement?

- What is the impact of DIS on reimbursement processes–are DIS being used to make reimbursement decisions for genomic assays?

### Strengths and limitations

The strength of our review is the extensive search of the literature and comprehensive categorization of clinical utility items reported. Using a broad search strategy reduced the probability that we missed any applicable studies. Due to resource limitations, we only included English language articles. As is typical with scoping reviews; we did not assess the quality of the included articles.

### Conclusion

The findings of this review provide a rigorous and comprehensive characterization of a new and expanding type of research in the field of genomic medicine and cancer care. Decision impact studies were first published 12 years ago, proliferated from breast cancer research to

other types of cancer and across numerous genomic assays, have been conducted in over 20 countries and report outcomes across 22 measures of clinical utility. These findings indicate that these studies are positioned to provide evidence for clinical and reimbursement decision-making. The results of this review provide important insights on these studies and can be leveraged by research endeavours that seek to further understand how decision impact studies are being used in decision-making for reimbursement for genomic assays in cancer care.

## Supporting information

**S1 Appendix. PRISMA-ScR checklist.** Completed PRISMA-ScR Checklist indicating page number in manuscript of relevant content.
(DOCX)

**S2 Appendix. Full electronic search strategy for Scopus database.**
(DOCX)

**S1 Table. Table of included studies.**
(DOCX)

## Acknowledgments

The authors would like to thank Helen Valkanas for her valuable assistance.

## Author Contributions

**Conceptualization:** Gillian Parker, Fiona A. Miller.

**Data curation:** Gillian Parker, Sarah Hunter, Samer Ghazi.

**Formal analysis:** Gillian Parker, Sarah Hunter, Fiona A. Miller.

**Funding acquisition:** Fiona A. Miller.

**Investigation:** Gillian Parker, Sarah Hunter, Samer Ghazi.

**Methodology:** Gillian Parker, Sarah Hunter, Fiona A. Miller.

**Project administration:** Gillian Parker.

**Supervision:** Fiona A. Miller.

**Visualization:** Gillian Parker, Sarah Hunter.

**Writing – original draft:** Gillian Parker.

**Writing – review & editing:** Gillian Parker, Sarah Hunter, Samer Ghazi, Robin Z. Hayeems, Francois Rousseau, Fiona A. Miller.

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
