## [Decision Letter · Decision Letter 0]

6 Oct 2022

PONE-D-22-22792Decision impact studies, evidence production and clinical utility in genomic testing in cancer care: A scoping reviewPLOS ONE

Dear Dr. Parker,

Thank you for submitting your manuscript to PLOS ONE. After careful consideration, we feel that it has merit but does not fully meet PLOS ONE’s publication criteria as it currently stands. Therefore, we invite you to submit a revised version of the manuscript that addresses the points raised during the review process.

We look forward to receiving your revised manuscript.

Kind regards,

Ramon Andrade De Mello, MD, PhD, FACP

Academic Editor

PLOS ONE

Journal Requirements:

2. We note that this manuscript is a systematic review or meta-analysis; our author guidelines therefore require that you use PRISMA guidance to help improve reporting quality of this type of study. Please upload copies of the completed PRISMA checklist as Supporting Information with a file name “PRISMA checklist”.

Reviewers' comments:

Reviewer's Responses to Questions

**Comments to the Author**

1. Is the manuscript technically sound, and do the data support the conclusions?

Reviewer #1: Yes

Reviewer #2: Yes

Reviewer #3: Yes

2. Has the statistical analysis been performed appropriately and rigorously? 

Reviewer #1: N/A

Reviewer #2: N/A

Reviewer #3: Yes

3. Have the authors made all data underlying the findings in their manuscript fully available?

Reviewer #1: Yes

Reviewer #2: Yes

Reviewer #3: Yes

4. Is the manuscript presented in an intelligible fashion and written in standard English?

Reviewer #1: Yes

Reviewer #2: Yes

Reviewer #3: Yes

5. Review Comments to the Author

Reviewer #1: Systematic review of a relevant and current topic. It objectively addresses the studies evaluated, including a significant number of studies published in the last decade, in the main search platforms. The results support the conclusions and favor an improvement in clinical practice. A pharmacoeconomics study would be appropriate, in order to corroborate the feasibility of using genomic tests.

Reviewer #2: Thank you for submit your article! It is interesting. I give you some comments:

TITLE: Please try to expressed in as few words as possible.

ABSTRACT: Try to synthetize the background section. Extend methods, you can briefly describe the inclusion and exclusion criteria using in the study selection and define your variables. How to categorize the types of clinical utility outcomes reported? Try to select only three or four keywords.

BACKGROUND: There are six paragraph, please try to resume in four paragraph. The first paragraph is good; in this you explain the utility of DIS. The second paragraph is regarding the use of genomic assays in cancer care. The third and the fourth paragraph could be summarize in only one paragraph. You should focus in the importance to assess clinical utility and the most accepted definition. Finally, the fifth and the sixth paragraph could be only one, try to explain the motivation for doing that research, show the knowledge gap that will lead into the importance of your study.

Study objectives: You repeat the importance of DIS. If the primary outcome is the impact of a genomic assay on treatment decisions or recommendations for a specific population of cancer patients. I am not sure if the primary objective should be only identify and characterize decision impact studies published for genomic assays in cancer care. Maybe you can change the primary objective for the second: describe the types of clinical utility outcomes reported as evidence of DIS in clinical and reimbursement decision-making.

METHODS: What type of studies were included? How many meta- analyses were conducted on that topic? Has the literature evolved significantly to justify an up- to- date meta- analysis? Are there any relevant studies missing in the already published meta- analyses?

RESULTS: The results, discussion and conclusions are good.

Reviewer #3: Decision impact studies are currently the best way to determine the most suitable treatment for each individual patient, according to their individual characteristics. A trend in oncology and precision medicine. The article was simple, easy to read and well written, it incorporated the most important studies of its category.

6. PLOS authors have the option to publish the peer review history of their article (what does this mean?). If published, this will include your full peer review and any attached files.

Reviewer #1: No

Reviewer #2: No

Reviewer #3: No

---

## [Author Response · Author response to Decision Letter 0]

11 Oct 2022

Dear Dr. De Mello,

Thank you for the invitation to revise and resubmit our manuscript. We have addressed the reviewer’s comments and believe that incorporating their feedback has enhanced the quality of our paper. Their comments were insightful, and we believe this review process has resulted in an improved manuscript. A detailed response to the comments may be found in the table below. 

Sincerely,

Gillian Parker

Reviewer Comment

Title

Please try to express in as few words as possible.

We have edited the title.

Abstract

2a Try to synthetize the background section. Extend methods, you can briefly describe the inclusion and exclusion criteria using in the study selection and define your variables. How to categorize the types of clinical utility outcomes reported? 

Thank you for this suggestion. We have edited and synthesized the abstract.

2b Try to select only three or four keywords. We have reduced the key words to six.

Background

3 There are six paragraphs, please try to resume in four paragraphs. 

The first paragraph is good; in this you explain the utility of DIS. The second paragraph is regarding the use of genomic assays in cancer care. The third and the fourth paragraph could be summarized in only one paragraph. You should focus on the importance to assess clinical utility and the most accepted definition. Finally, the fifth and the sixth paragraph could be only one, try to explain the motivation for doing that research, show the knowledge gap that will lead into the importance of your study.

Thank you for this suggestion. We have edited and re-organized the background section.

Study Objectives

4 You repeat the importance of DIS. If the primary outcome is the impact of a genomic assay on treatment decisions or recommendations for a specific population of cancer patients. I am not sure if the primary objective should be only identify and characterize decision impact studies published for genomic assays in cancer care. Maybe you can change the primary objective for the second: describe the types of clinical utility outcomes reported as evidence of DIS in clinical and reimbursement decision-making.

Thank you for this observation. We have removed the duplicated statement. We decided to conduct a scoping review because there is no published synthesis of the literature on decision impact studies. A scoping review is foundational study to establish the nature and scope of a body of research. 

We have clarified this objective in the manuscript. 

Methods

5a What type of studies were included? 

Thank you for identifying this omission. We have added detail regarding ‘type of studies’ included in this review.

5b How many meta- analyses were conducted on that topic? Has the literature evolved significantly to justify an up- to- date meta- analysis? Are there any relevant studies missing in the already published meta- analyses?

Thank you for this inquiry. While two meta-analyzes on tangentially related topics (comparing radiomic MRI and RS (2021) and use of Oncotype Dx (2015)) have been published and cite some DIS studies in their included studies, to our knowledge, no reviews have been conducted of decision impact studies. This fact motivated us to conduct this review. Our intention is to provide a comprehensive synthesis of DIS and contribute to foundational knowledge in this emerging field of research. 

We have explicitly stated the gap in knowledge synthesis in this field in the objectives statement.

---

## [Decision Letter · Decision Letter 1]

4 Jan 2023

Decision impact studies, evidence of clinical utility for genomic assays in cancer: A scoping review

PONE-D-22-22792R1

Dear Dr. Miller,

We’re pleased to inform you that your manuscript has been judged scientifically suitable for publication and will be formally accepted for publication once it meets all outstanding technical requirements.

Kind regards,

Meng Li

Academic Editor

PLOS ONE

Additional Editor Comments (optional):

Reviewers' comments:

Reviewer's Responses to Questions

**Comments to the Author**

1. If the authors have adequately addressed your comments raised in a previous round of review and you feel that this manuscript is now acceptable for publication, you may indicate that here to bypass the “Comments to the Author” section, enter your conflict of interest statement in the “Confidential to Editor” section, and submit your "Accept" recommendation.

Reviewer #2: All comments have been addressed

2. Is the manuscript technically sound, and do the data support the conclusions?

Reviewer #2: Yes

3. Has the statistical analysis been performed appropriately and rigorously? 

Reviewer #2: N/A

4. Have the authors made all data underlying the findings in their manuscript fully available?

Reviewer #2: (No Response)

5. Is the manuscript presented in an intelligible fashion and written in standard English?

Reviewer #2: (No Response)

6. Review Comments to the Author

Reviewer #2: ABSTRACT:

Try to select only three or four keywords.

BACKGROUND

Please synthetize the last paragraph. Try to focus in the importance to assess clinical utility, resume the most relevant previous studies and show the knowledge gap that will lead into the importance of your study.

Study objectives: The first sentence is not appropriately for this section, you can include this in the last paragraph of background section. I understand that the primary outcome identify and characterize decision impact studies in genomic cancer testing.

7. PLOS authors have the option to publish the peer review history of their article (what does this mean?). If published, this will include your full peer review and any attached files.

Reviewer #2: **Yes: **Katia Roque

---

## [Editor Report · Acceptance letter]

9 Jan 2023

PONE-D-22-22792R1 

Decision impact studies, evidence of clinical utility for genomic assays in cancer: A scoping review 

Dear Dr. Miller:

I'm pleased to inform you that your manuscript has been deemed suitable for publication in PLOS ONE. Congratulations! Your manuscript is now with our production department. 

Kind regards, 

on behalf of

Dr. Meng Li 

Academic Editor

PLOS ONE